

# Registered report: androgen receptor splice variants determine taxane sensitivity in prostate cancer

Xiaochuan Shan[1], Gwenn Danet-Desnoyers[1], Juan José Fung[2], Alan H. Kosaka[2], Fraser Tan[3], Nicole Perfito[3], Joelle Lomax[3] and Elizabeth Iorns[3]

[1] Stem Cell and Xenograft Core, Perelman School of Medicine, University of Pennsylvania, Philadelphia, PA, Unites States
[2] ProNovus Bioscience LLC, Mountain View, CA, United States
[3] Science Exchange and The Prostate Cancer Foundation-Movember Foundation Reproducibility Initiative, Palo Alto, CA, United States

## ABSTRACT

The Prostate Cancer Foundation-Movember Foundation Reproducibility Initiative seeks to address growing concerns about reproducibility in scientific research by conducting replications of recent papers in the field of prostate cancer. This Registered Report describes the proposed replication plan of key experiments from "Androgen Receptor Splice Variants Determine Taxane Sensitivity in Prostate Cancer" by Thadani-Mulero and colleagues (*2014*) published in *Cancer Research* in 2014. The experiment that will be replicated is reported in Fig. 6A. Thadani-Mulero and colleagues generated xenografts from two prostate cancer cell lines; LuCaP 86.2, which expresses predominantly the ARv567 splice variant of the androgen receptor (AR), and LuCaP 23.1, which expresses the full length AR as well as the ARv7 variant. Treatment of the tumors with the taxane docetaxel showed that the drug inhibited tumor growth of the LuCaP 86.2 cells but not of the LuCaP 23.1 cells, indicating that expression of splice variants of the AR can affect sensitivity to docetaxel. The Prostate Cancer Foundation-Movember Foundation Reproducibility Initiative is a collaboration between the Prostate Cancer Foundation, the Movember Foundation and Science Exchange, and the results of the replications will be published by *PeerJ*.

## INTRODUCTION

Prostate cancer is one of the most prevalent forms of cancer occurring in men, and its progression is dependent upon the androgen receptor (AR) signaling pathway. Initial treatment by androgen deprivation therapy (ADT) can prove efficacious; however, relapse is common, resulting in castration-resistant prostate cancer (CRPC). Despite low levels of androgens, AR signaling remains active in CRPC through a variety of mechanisms. These include amplification of the AR locus, mutations in the AR leading to increased and promiscuous ligand sensitivity, or ligand-independent activation, among others (*Ferraldeschi et al., 2014*). Additionally, alternatively-spliced variants of the AR that lead to

Corresponding author
Fraser Tan,
fraser@scienceexchange.com

protein truncation and cause loss of the ligand binding domain can result in constitutively active forms of the receptor (*Ware et al., 2014*).

Once ADT fails, the standard second-line treatment for CRPC is the anti-mitotic drug docetaxel, a taxane that stabilizes microtubules. This prevents their dynamic assembly and disassembly, which results in cellular apoptosis. Docetaxel is also thought to interrupt the microtubule-based translocation of the AR nuclear receptor itself (*Ferraldeschi et al., 2014*; *Martin & Kyprianou, 2015*). Not all CRPC responds to docetaxel, however; different splice variants associated with CRPC and metastasis can result in differential sensitivity to taxanes (*Lu, Van der Steen & Tindall, 2015*; *Sprenger & Plymate, 2014*).

In their 2014 *Cancer Research* paper, Thadani-Mulero and colleagues explored how two common AR splice variants, ARv567 and ARv7, responded to treatment with docetaxel. Microtubule sedimentation assays showed that the ARv567 variant heavily associated with microtubules, while the ARv7 variant did not. They confirmed this finding *in vitro* by treating cells with microtubule stabilization and destabilization agents, and observed significant impairment of nuclear accumulation of ARv567, but not ARv7.

In Fig. 6A, they performed a xenograft growth assay using two different prostate cancer xenograft lines; LuCaP 86.2, which expresses predominantly ARv567, and LuCaP 23.1, which co-expresses wild-type AR and ARv7. They showed that treatment of the LuCaP 86.2 tumors with docetaxel significantly reduced tumor growth, while treatment of the LuCaP 23.1 tumors did not. This key experiment will be replicated in Protocol 1. In Protocol 2, expression of the AR variants in each tumor type will be confirmed by Western Blot.

Previous work by Brubaker and colleagues had demonstrated that treatment of subcutaneous LuCaP 23.1 xenografts with docetaxel did decrease tumor volume (*Brubaker et al., 2006*), a finding not recapitulated by Thadani-Mulero and colleagues. However, Martin and colleagues presented data corroborating the finding that treatment with a taxane reduced nuclear translocation of the full length AR but not of AR variants. They also showed that treatment of 22Rv1 prostate cell xenografts with docetaxel did not significantly reduce cell growth. Like LuCaP 23.1 cells, 22Rv1 cells express both full length AR and ARv7 (*Martin et al., 2014*).

## MATERIALS & METHODS

Unless otherwise noted, all protocol information was derived from the original paper, references from the original paper, or information obtained directly from the authors. An asterisk (*) indicates data or information provided by the PCFMFRI core team. A hashtag (#) indicates information provided by the replicating lab. All references to Figures are in reference to the original study.

### Protocol 1: response of xenograft tumors derived from LuCaP 86.2 and LuCaP 23.1 prostate cancer cells to treatment with docetaxel

This protocol describes how to generate xenograft tumors derived from LuCaP 86.2 prostate cancer cells, which harbor the ARv567 androgen receptor (AR) truncation mutant, and LuCaP 23.1 prostate cancer cells, which harbor predominantly wild-type AR and the ARv7 truncation mutant. Mice bearing these xenograft tumors are treated with docetaxel and tumor volume is measured over the course of 8 weeks, as seen in Fig. 6A.

### Sampling

- The experiment will use at least 11 mice per group for a final power of at least 84.95%.
  ○ See Power Calculations section for details
- Each experiment has two cohorts, each of which is split into two groups (4 groups total):
  ○ Cohort 1, Group 1: Mice bearing LuCaP 86.2 prostate cancer xenografts, uninjected
    ∗ $N = 11$ mice
  ○ Cohort 1, Group 2: Mice bearing LuCaP 86.2 prostate cancer xenografts, treated with docetaxel
    ∗ $N = 11$ mice
  ○ Cohort 2, Group 1: Mice bearing LuCaP 23.1 prostate cancer xenografts, uninjected
    ∗ $N = 11$ mice
  ○ Cohort 2, Group 2: Mice bearing LuCaP 23.1 prostate cancer xenografts, treated with docetaxel
    ∗ $N = 11$ mice
  ○ In total, 24 mice with LuCaP 86.2 tumors and 24 mice with LuCaP 23.1 tumors are generated.

### Materials and reagents

| Reagent | Type | Manufacturer | Catalog # | Comments |
|---|---|---|---|---|
| LuCaP 86.2 | Tumor tissue | Shared by original authors | | |
| LuCaP 23.1 | Tumor tissue | Shared by original authors | | |
| Male CB17 SCID mice | Mice | Charles River | Strain Code 236 | |
| Docetaxel | Drug | LC Laboratories | D-1000 | |

### Procedure

Notes:

- Fresh tumor tissue, shipped overnight from the original authors, will be used; tissue will not be frozen.
- Tumor tissue will be screened with a Rodent Pathogen Panel to confirm no pathogens are present.
- Information in this protocol is derived from *Mostaghel et al. (2011)*, *Wu et al. (2006)* and *Zhang et al. (2011)*.
- Docetaxel is prepared fresh on the day it will be used in 13% ethanol/0.9% NaCl.
- * Experimenters should be blinded to the treatment of the mice.

1. Mince fresh LuCaP 86.2 and LuCaP 23.1 tumor tissue into small fragments 20 mm$^3$ in size.
   (a) If multiple tumors are provided, optimally only a single tumor will be used for implantation. If multiple tumors are needed to generate enough fragments for implantation, all tumors will be minced to appropriate sizes and half the mice will receive tissue from one tumor, while the other half receive tissue from the other tumor. The donating tumor will be recorded for each mouse.

2. Subcutaneously implant non-castrated 6–8 week old male SCID mice on the right flank-shoulder area with tumor fragments 20 mm$^3$ in size.
   (a) Generate 24 mice bearing LuCaP 86.2 tumors.
   (b) Generate 24 mice bearing LuCaP 23.1 tumors.

3. Let tumors grow to 100 mm$^3$ prior to the start of treatment.
   (a) Measure tumor volume twice weekly.
      i. Volume = length × width × height × 0.5236
   (b) Growth characteristics can be variable; time to enrollment may be between one to two months.
   (c) Note: treatment initiation will not be synchronized across tumors.

4. Randomly assign mice to the treatment group and the control group. Once tumors reach 100 mm$^3$, non-control mice are treated by intraperitoneal injections every other week for eight weeks.
   (a) $^\#$ Animals are randomized according to a stratified randomization procedure balanced for final tumor volume and spread of tumor volume.
   (b) Control mice receive no injections.
      i. Note: This information is based on communication from the original authors.
   (c) Treated mice receive 10 mg/kg docetaxel in 400 μL 13% ethanol/0.9% NaCl per injection.
      i. Note: This information is based on communication from the original authors.

5. Measure tumor volume twice weekly for duration of experiment.

6. Continue treatment for 8 weeks.
   (a) * Inject mice in weeks 1, 3, 5, and 7.
   (b) Euthanize animals when they display one or more of the following conditions:
      i. Tumor volume exceeds 1,000 mm$^3$
      ii. >20% body weight loss
      iii. Animals become compromised (hunched posture, piloerected, rapid respiration, lethargic)

7. In Week 8, sacrifice mice.
   (a) For each untreated group, randomly select three mice and harvest tumor tissue (6 tumors total; 3 uninjected LuCaP 86.2, 3 uninjected LuCaP 23.1).
      i. $^\#$ Snap freeze tumor tissue in liquid nitrogen and stored at −80 °C until ready for use.

### Deliverables

- Data to be collected:
  ○ All mouse health records (age, gender, date of implantation, size of injected tissue fragment, treatment regimen, date and cause of euthanasia)
  ○ Raw measurements of tumor size and calculated tumor volume for each mouse for all weeks measured
  ○ Graph of average tumor size per group each week, as seen in Fig. 6A.

   * To generate weekly mean measurements, first average the two measurements for that week for each tumor. Then average the averaged tumor measurements within each group to generate a group mean tumor volume.

  ◦ * Graph of median tumor size per group each week.

   * Unlike the average measurements, do not combine weekly tumor size measurements when calculating the media tumor size per group each week.

- Sample delivered for further analysis:
  ◦ Snap frozen control-group tumor tissues ready for use in Protocol 2.

### Confirmatory analysis plan

- Statistical Analysis of the Replication Data:
  ◦ Note: At the time of analysis we will perform the Shapiro–Wilk test and generate a quantile–quantile plot to assess the normality of the data. We will also perform Levene's test to assess homoscedasticity. If the data appears skewed we will perform the appropriate transformation in order to proceed with the proposed statistical analysis. If this is not possible we will perform the planned comparison using the appropriate nonparametric test.

  ◦ One-way ANOVA on Week 8 time points followed by Bonferroni corrected $t$-tests comparing:

   * LuCaP86.2 untreated vs. LuCaP 86.2 treated with docetaxel
    · As performed by the original authors

- Meta-analysis of original and replication attempt effect sizes:
  ◦ This replication attempt will perform the statistical analysis listed above, compute the effects sizes, compare them against the reported effect size in the original paper and use a meta-analytic approach to combine the original and replication effects, which will be presented as a forest plot.

- Additional Analysis of the Replication Data:
  ◦ Two-way ANOVA ($2 \times 2$) assessing area under the curve followed by Bonferroni corrected $t$-test comparisons:

   * LuCaP86.2 untreated vs. LuCaP 86.2 treated with docetaxel
   * LuCaP 86.2 treated with docetaxel vs. LuCaP 23.1 treated with docetaxel

### Known differences from the original study

All known differences in reagents and supplies are listed in the materials and reagents section above, with the originally used item listed in the comments section. All differences have the same capabilities as the original and are not predicted to alter experimental outcome.

### Provisions for quality control

All data obtained from the experiment—raw data, data analysis, control data and quality control data—will be made publicly available, either in the published manuscript or as an open access dataset available on the Open Science Framework (https://osf.io/gkd2u/).

- Results of the Rodent Pathogen Panel screening

## Protocol 2: Western blot analysis confirming expression of AR truncation mutants in xenograft tumor tissue

This protocol describes how to assess levels of protein expression of AR truncation mutants in xenograft tumor tissue from Protocol 1, as seen in Figure 6B. This is a quality control experiment to confirm the presence of the expected AR truncation mutants in each xenograft cell type. LuCaP23.1 tissue expresses both the full-length AR and Arv7, while LuCaP86.2 tissue expressed some full-length AR but predominantly Arv567.

### Sampling

- The experiment has two cohorts, derived from Protocol 1;
  - Cohort 1: 3 random tumors derived from uninjected LuCaP 86.2 prostate cancer xenografts
  - Cohort 2: 3 random tumors derived from uninjected LuCaP 23.1 prostate cancer xenografts
- Each sample will be probed with antibodies for the following targets:
  - ARN20
    - ∗ Detects the full length AR as well as the AR567 splice variant, which runs slightly faster than the full length AR
  - Arv567es
    - ∗ Detects the Arv567 splice variant
  - Arv7
    - ∗ Detects the ARv7 splice variant
  - Beta-Actin
    - ∗ Housekeeping control
- The experiment will be performed on 3 samples per cohort.
  - This experiment is exploratory in nature, and thus no power calculations are necessary.

### Materials and reagents

| Reagent | Type | Manufacturer | Catalog # | Comments |
| --- | --- | --- | --- | --- |
| Protease inhibitor, complete, mini, EDTA-free | Protease inhibitor | Roche | 04693159001 | Original unspecified |
| Rabbit $\alpha$ ARN20 | Antibody | Santa Cruz | sc-816 | 1:200 |
| Mouse monoclonal IgG$_{2A}\alpha$ ARv7 | Antibody | Precision Antibody | AG10008 | 2 $\mu$ g/$\mu$ L |
| HRP-conjugated mouse monoclonal IgG$_1\alpha$ beta-actin | Antibody | Sigma Aldrich | A3854 | 1:25,000 |
| Goat anti-mouse IgG-HRP | Antibody | Bio-Rad | 172-1011 | 1:10,000 |
| Goat anti-rabbit IgG-HRP | Antibody | Santa Cruz | Sc-2030 | 1:2,000 |
| Bradford Protein Assay kit, with BSA standards | Protein Assay | Bio-Rad | 500-0002 | |
| TNES buffer (will be made in-house): Tris, NaCl, EDTA, Nonidet P-40 | Buffer | Sigma | various | |
| Precision Plus Protein All Blue Standards | Protein Ladder | Bio-Rad | 161-0373 | |
| 1-Step TMB blot solution | Western detection | Life Technologies | 34018 | |
| Rabbit monoclonal $\alpha$ ARv567es | Antibody | Abcam | ab200827 | Recommended by original authors |

### Procedure

Notes:

- Information in this protocol obtained from Darshan and colleagues (*2011*) and from the replicating lab.
- This protocol will use snap frozen tumor tissue generated in Protocol 1.

1. Preparation of samples: Lyse cells in TNES buffer.
   (a) TNES buffer: 50 mM Tris pH 6.0, 100 mM NaCl, 2 mM EDTA, 1% Nonidet P-40, 1X Protease Inhibitor
   (b) [#] Frozen tissue will be homogenized/sonicated on ice, then spun down at 13,000 rpm at 4 °C.
   (c) [#] Supernatant is diluted 1:1 to perform a Bradford Protein Assay according to manufacturer's protocol.
2. Separate [#]50 μg per well of protein on a 10% SDS-PAGE gel.
3. Transfer protein to [#] nitrocellulose membrane.
4. [#] Block with 1% non-fat dry milk in PBS/0.05% Tween-20 for 60 min at room temperature (RT).
   (a) Wash 3 × 5 min in PBS/0.05% Tween-20.
5. [#] Incubate primary antibodies (at denoted concentration/dilution) in PBS/0.05% Tween-20 overnight at 4 °C.
   (a) $\alpha$ ARN20: 1:200
   (b) $\alpha$ Arv567es; 1:3,000
   (c) $\alpha$ ARv7: 2 μg/μL
   (d) $\alpha$ Beta-actin: 1:25,000
6. [#] Wash 3 × 5 min in PBS/0.05% Tween-20.
7. [#] Incubate with secondary antibodies for 60 min at RT.
   (a) Goat-anti-mouse IgG-HRP; 1:10,000
   (b) Goat anti-rabbit IgG-HRP; 1:2,000
8. [#] For detection:
   (a) Incubate membrane with 5 mL of TMB detection reagent. Stop with distilled water upon achieving desired color development
9. [#] Image using an 8.0 megapixel digital camera.

### Deliverables:

- Data to be collected:
  ○ All raw gel images including ladder marker indication

### Confirmatory analysis plan

- None necessary

### Known differences from the original study

All known differences in reagents and supplies are listed in the materials and reagents section above, with the originally used item listed in the comments section. All differences have the same capabilities as the original and are not predicted to alter experimental outcome.

- The replication will exclude the LuCaP35 cell line from this experiment.
- The replicating lab will use their in-house Western blot protocol with colorimetric detection. This replaces the LiCOR Odyssey detection used by the original lab.
- At the recommendation of the original authors, we will use an Arv567es specific antibody (Abcam 200827) to detect the V567es variant protein instead of relying upon the ARN20 antibody to detect both full length ant v567es proteins.

### Provisions for quality control

All data obtained from the experiment—raw data, data analysis, control data and quality control data—will be made publicly available, either in the published manuscript or as an open access dataset available on the Open Science Framework (https://osf.io/gkd2u/).

## POWER CALCULATIONS

For details on power calculations, please see https://osf.io/grk54/?view_only=685e665bb55a47c2aeec346007a228c5.

## Protocol 1

### Summary of original data

| Figure 6A | | Mean tumor volume | SEM | SD | N |
|---|---|---|---|---|---|
| | Week 1 | 122.98 | 68.94 | 267.02 | 15 |
| | Week 2 | 162.73 | 38.51 | 149.14 | 15 |
| | Week 3 | 279.50 | 23.60 | 91.41 | 15 |
| LuCaP 86.2; control | Week 4 | 347.83 | 34.78 | 134.71 | 15 |
| | Week 5 | 501.86 | 80.75 | 312.72 | 15 |
| | Week 6 | 627.33 | 131.68 | 509.98 | 15 |
| | Week 7 | 914.29 | 96.89 | 375.27 | 15 |
| | Week 8 | 977.64 | 111.80 | 433.00 | 15 |
| | Week 1 | 113.04 | 0.00 | 0.00 | 15 |
| | Week 2 | 159.01 | 33.54 | 129.90 | 15 |
| | Week 3 | 114.29 | 0.00 | 0.00 | 15 |
| LuCaP 86.2; docetaxel | Week 4 | 121.74 | 28.57 | 110.66 | 15 |
| | Week 5 | 125.47 | 32.30 | 125.09 | 15 |
| | Week 6 | 80.75 | 0.00 | 0.00 | 15 |
| | Week 7 | 74.53 | 0.00 | 0.00 | 15 |
| | Week 8 | 78.26 | 0.00 | 0.00 | 15 |

| Figure 6A | | Mean tumor volume | SEM | SD | N |
|---|---|---|---|---|---|
| LuCaP 23.1; control | Week 1 | 160.25 | 26.09 | 101.03 | 15 |
| | Week 2 | 289.44 | 49.69 | 192.44 | 15 |
| | Week 3 | 462.11 | 103.11 | 399.33 | 15 |
| | Week 4 | 602.48 | 139.13 | 538.85 | 15 |
| | Week 5 | 555.28 | 147.83 | 572.53 | 15 |
| | Week 6 | 488.20 | 98.14 | 380.08 | 15 |
| | Week 7 | 750.31 | 227.33 | 880.44 | 15 |
| | Week 8 | – | – | – | – |
| LuCaP 23.1; docetaxel | Week 1 | 145.34 | 52.17 | 202.07 | 15 |
| | Week 2 | 237.27 | 29.81 | 115.47 | 15 |
| | Week 3 | 375.16 | 33.54 | 129.90 | 15 |
| | Week 4 | 489.44 | 40.99 | 158.77 | 15 |
| | Week 5 | 586.34 | 126.71 | 490.74 | 15 |
| | Week 6 | 655.90 | 101.86 | 394.51 | 15 |
| | Week 7 | 654.66 | 129.19 | 500.36 | 15 |
| | Week 8 | 655.90 | 185.09 | 716.86 | 15 |

- Stdev was calculated using formula $SD = SEM*(SQRT\ n)$.

### Test family
- One-way ANOVA on Week 8 time points followed by Bonferroni corrected $t$-tests comparing:
  - LuCaP86.2 untreated vs. LuCaP 86.2 treated with docetaxel
    - ∗ As performed by the original authors
- Additional analyses:
  - Two-way ANOVA ($2 \times 2$) assessing Area Under the Curve followed by Bonferroni corrected $t$-test comparisons:
    - ∗ LuCaP86.2 untreated vs. LuCaP 86.2 treated with docetaxel
    - ∗ LuCaP 86.2 treated with docetaxel vs. LuCaP 23.1 treated with docetaxel

### Power calculations
- Power calculations were performed using R (*R Core Team, 2014*), GraphPad PRISM v6 for Mac and G*Power (version 3.1.7) (*Faul et al., 2007*)
- One way ANOVA as originally performed.
  - Note: This excludes the LuCaP 23.1 cohort that is missing an 8 week time point.

| One way ANOVA; $\alpha = 0.05$, 3 groups. | | | | |
|---|---|---|---|---|
| F(2,42) | Partial eta$^2$ | Effect size $f$ | A priori power | Total $n$ |
| 13.32 | 0.388112 | 0.7964208 | 85.96% | 21 |

| Bonferroni corrected $t$-test; $\alpha = 0.05$. | | | | |
|---|---|---|---|---|
| Group 1 | Group 2 | Effect size $d$ | A priori power | N per group |
| LuCaP86.2 untreated | LuCaP 86.2 treated with docetaxel | 2.93742 | 92.99% | 4 |

- Two-way ANOVA on area under the curve followed by Bonferroni corrected $t$-test comparisons
  - Calculated with R (*R Core Team, 2014*).

| Area under the curve | Mean | AUC SD | N |
|---|---|---|---|
| LuCaP 86.2 TXT 5 mg/mL | 695.0417 | 365.645 | 15 |
| LuCaP 23.1 | 2,852.795 | 2,573.968 | 15 |
| LuCaP 86.2 | 2,437.887 | 1,519.118 | 15 |
| LuCaP 23.1 TXT 5 mg/kg | 2,744.103 | 1,640.595 | 15 |

| Two way ANOVA; $\alpha = 0.05$, 4 groups. | | | | |
|---|---|---|---|---|
| F(3,56) | Partial eta$^2$ | Effect size $f$ | A priori power | Total $n$ |
| 5.18 | 0.217057 | 0.526528 | 80.61% | 44 |

| Bonferroni corrected $t$-test. Power calculations; $\alpha = 0.025$. | | | | |
|---|---|---|---|---|
| Group 1 | Group 2 | Effect size $d$ | A priori power | N per group |
| LuCaP 86.2 untreated | LuCaP 86.2 treated with docetaxel | 1.57744 | 84.95% | 10 |

| Sensitivity calculations; $\alpha = 0.025$. | | | | |
|---|---|---|---|---|
| Group 1 | Group 2 | Detectable effect size $d$ | Power | N per group |
| LuCaP 23.1 untreated | LuCaP 23.1 treated with docetaxel | 1.4841766 | 80.00% | 10 |

## Protocol 2

- No power calculations necessary.

## ACKNOWLEDGEMENTS

The PCFMFRI core team consists of Elizabeth Iorns, Fraser Tan, Joelle Lomax and Nicole Perfito. The PCFMFRI core team would like to thank the original authors, in particular Dr. Paraskevi Giannakakou, Dr. Stephen R. Plymate, Dr. Robert Vessella and Dr. Eva Corey, for generously sharing critical information as well as reagents to ensure the fidelity and quality of this replication attempt.

### Funding

The Prostate Cancer Foundation Movember Foundation Reproducibility Initiative is funded by the Prostate Cancer Foundation and the Movember Foundation. The funders had no role in study design, data collection and analysis, decision to publish, or preparation of the manuscript.

## Grant Disclosures

The following grant information was disclosed by the authors:
Prostate Cancer Foundation and the Movember Foundation.

## Competing Interests

Elizabeth Iorns, Fraser Tan, Joelle Lomax and Nicole Perfito are employed by and hold shares in Science Exchange Inc. The experiments presented in this manuscript will be conducted by XC and GDD at the Stem Cell and Xenograft Core and by JF and AK at ProNovus Biosciences LLC, which are Science Exchange labs.

## Author Contributions

- Xiaochuan Shan, Gwenn Danet-Desnoyers, Juan José Fung, Alan H. Kosaka and Elizabeth Iorns conceived and designed the experiments, wrote the paper.
- Fraser Tan conceived and designed the experiments, reviewed drafts of the paper.
- Nicole Perfito and Joelle Lomax conceived and designed the experiments, wrote the paper, reviewed drafts of the paper.

## Data Availability

The following information was supplied regarding the availability of data:
Open Science Framework: https://osf.io/gkd2u/.

## Supplemental Information

Supplemental information for this article can be found online at http://dx.doi.org/10.7717/peerj.1232#supplemental-information.

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
