# Peer review of "Registered report: androgen receptor splice variants determine taxane sensitivity in prostate cancer"

_PeerJ, doi:10.7717/peerj.1232_

## Round 0.1 · original submission · Major Revisions

Please address the concerns of the reviewers in your revised document. With respect to the comments on data analysis by average versus median tumor size, you may want to consider including both procedures in your protocol. Let us know if you have questions during the revision process.

Reviewer 1 ·

Basic reporting

1. Please explain why the correspondence person is not in the authorship.
2. The source of LuCa P23.1 in the study of Brubaker et al., 2006 is not explicitly described. It is possible that the tumor may not be the same one described in the study of Thadani-Mulero et al., 2014, in which the source of LuCaP 23.1 was referred to Clin Cancer Res 2006;12:6153–60.
3. Both papers of S. K. Martin et al are not cited with complete information.

Experimental design

1. Please describe how the frozen tumor will be revived and expanded for this study. For example, if the starting frozen tumor will be chopped and transplanted into several mice, will only one tumor will be selected for transplantation in study cohort? If more than one are needed for sufficient material, what is the randomization procedure?
2. The take rate of human tumor xenograft in immunocompromised mice can be highly variable between experiments, and 80% is already regarded as very good. The mouse cohort should be expanded to buffer lower tumor take rate in addition to unexpected death.
3. For human tumor xenograft, the initial condition of tumor growth may be highly heterogeneous. Therefore, the time to reach 100 mm3 (about 6 mm x 6 mm) can also be variable among individual tumors, resulting in the difficulty in the synchronization of treatment schedule. A plan in respond to such situation should be described. For example, the mouse cohort can be further expand to allow sufficient number of tumor to reach the size range of 60 mm3 (about 5 mm x 5 mm) to 170 mm3 (about 7 mm x 7 mm) for initiating the treatment. Alternatively, the tumor can be monitored to start treatment individually without synchronization, but all the treatments need to be initiated in a time range (e.g. 1 to 2 month). The authors seem to use the latter. If so, please specify the treatments (including vehicle control, see point 3 below) will not be synchronized. In either case, the randomization procedure should be noted.
4. The control groups should be treated with vehicle solution (13% ethanol/0.9% NaCl), instead of being left without treatment.
5. In the study of Thadani-Mulero et al., 2014, mice was treated with either vehicle or docetaxel 5 or 20-mg/kg i.p. weekly. In Protocol 1, mice will be injected i.p. 10 mg/kg of docetaxel in week 1, 3, 5, 7. Please explain the difference.
6. Growth heterogeneity- particularly for human tumor xenografts- is unavoidable. Using group median, rather than the group average, tumor size for calculation of endpoints can track better with the individual data and reduce the skewing by outlier (for detail, see Hollingshead MG, JNCI 2008, 100:1500). The authors may consider to report data using both average+/-SD (or SEM) and median+/-interquartile range.

Validity of the findings

Not applied.

Additional comments

The heterogeneity in tumor-take rate and intra-group growth is the most profound factor affecting the reproducibility of preclinical studies using human tumor xenograft (Hollingshead MG, JNCI 2008, 100:1500). Moreover, heterogeneity of data increases the chance of misuse statistic criteria, giving biased output. The authors have used power calculator to estimate the minimally required number of the mice for this study. Here are a few technical suggestions that may help to further reduce heterogeneity in this protocol. If possible, the published data to be compared with the results of this study should be examined to see if these technical concerns had also been addressed in their protocols.

·

Basic reporting

no comments

Experimental design

Dose of docetaxel should be 20mg/kg not 10 mg/kg.
To identify ARv567es use the ARv567es antibody from ABcam

Validity of the findings

no comment

Additional comments

please correct docetaxel dose and use ABcan v567es antibody

---

## Round 0.2 · Minor Revisions

Please attend to the final suggested revisions from Reviewer 1, and the submission can then be accepted without further review.

Reviewer 1 ·

Basic reporting

The authors have addressed all my comments in the revised version. A few minor points are listed in the "General Comments" section below.

Experimental design

No Comments

Validity of the findings

No Comments

Additional comments

1. Please note explicitly in the current protocol that fresh tumors, shipped overnight from the original authors' lab, will be used, rather than the frozen material. Therefore, no tumor revival is needed. The procedure when the second tumor is needed also has to be described.

2. In the result report, please note that the original authors did not evaluate the effect of vehicle solution.

---

## Round 0.3 · accepted · Accept

Please proceed with the experiments as outlined in this Registered Report. We look forward to receiving the resulting manuscript for this Replication Study.